# OpenReview forum: "SYN-TITAN: Synthetic Tabular Intelligence using Transformers and Adversarial Networks"
_ICLR.cc/2026/Conference — ICLR 2026 Conference Desk Rejected Submission_

### Official Review · Reviewer_aoDd · 2025-10-18

**Soundness:** 1
**Presentation:** 1
**Contribution:** 1
**Rating:** 0
**Confidence:** 5

**Summary:**

This paper, SYN-TITAN: Synthetic Tabular Intelligence using Transformers and Adversarial Networks, proposes a hybrid framework that integrates large language models (LLMs) with generative adversarial networks (GANs) to improve synthetic tabular data generation.
The system uses an LLM for three stages: pre-processing, generation, and post-processing. Experiments are conducted only on the UCI Adult Census Income dataset, comparing SYN-TITAN with CTGAN and Enhanced CTGAN. The reported results suggest minor improvements in logical consistency and correlation preservation, though the evaluation remains narrow in scope.

**Strengths:**

S1. Addresses an important problem of privacy-preserving tabular data synthesis.

S2. The motivation for integrating LLM reasoning with GAN generation is conceptually interesting.

S3. The paper is easy to follow.

**Weaknesses:**

W1. Shallow Technical Contribution:
The main novelty is a heuristic prompt that asks an LLM to list key features and domain constraints, followed by a simple filtering step to remove rows that violate those rules. There is no principled algorithmic innovation—the LLM’s outputs are used as static hints rather than being integrated into the training objective. The paper lacks ablations on prompt design, alternative formulations, or theoretical justification for why the chosen prompting strategy is optimal or robust.

W2. Weak Engagement with Related Work:
The paper positions itself mainly against CTGAN and Enhanced CTGAN, ignoring a large body of recent literature on tabular synthesis and LLM-based tabular modeling. Important works include:

- Tabular Data Synthesis with Generative Adversarial Networks: Design Space and Optimizations (VLDB J., 2024).
- Transformers for Tabular Data Representation: A Survey of Models and Applications. (Trans. Assoc. Comput. Linguistics 2023)
- More recent models such as CTAB-GAN+ (https://arxiv.org/abs/2204.00401), TabDDPM (https://arxiv.org/abs/2209.15421), and other diffusion- or LLM-based synthesizers that already provide superior fidelity and controllability.

Without comparing against these, the claimed novelty and improvement are unsubstantiated.

W3. Extremely Limited Evaluation:
The paper uses only one dataset (Adult Census) and one weak baseline (CTGAN). This cannot support any claim of generalization or scalability. No tests on complex or domain-specific datasets (e.g., healthcare, finance) where LLM prior knowledge may fail. No benchmarking with standard suites like SDGym or evaluation on downstream ML utility (e.g., TSTR). Privacy is mentioned but not implemented—“differential privacy” is only stated as planned.

W4. Overstated Claims and Missing Analysis:
The results show modest or negligible gains over CTGAN, and sometimes even worse distributional metrics. Logical consistency is enforced by post-hoc deletion, not by integrated learning. No analysis of prompt sensitivity, failure cases, or computational cost is provided. The method’s reliance on LLM priors also raises questions about hallucinated or domain-biased constraints, which are not examined.

W5. Overall Assessment:
Despite an appealing idea (LLM-guided GAN synthesis), the execution is methodologically weak, empirically narrow, and not competitive with existing baselines. The work lacks depth, novelty, and convincing evidence to merit acceptance at ICLR.

**Questions:**

Please see the above weaknesses W1--W4.

---

### Official Review · Reviewer_wr1n · 2025-10-26

**Soundness:** 1
**Presentation:** 1
**Contribution:** 1
**Rating:** 0
**Confidence:** 4

**Summary:**

The paper presents a GAN based method using LLM to generate synthetic data.

**Strengths:**

None.

**Weaknesses:**

The paper is clearly not qualified for a scientific publication. It is more like a unfinished technical report. Using bullet points throughout the presentation. There are a lot of work in this field, while the author only bechmark one of them, CTGAN in section 5.2. I suggest the author first do a carefully literature review, and try different methods before deploying yours (e.g. tabsyn, tabdppm, beGreaT, etc.).

**Questions:**

None

---

### Official Review · Reviewer_Zpno · 2025-10-27

**Soundness:** 1
**Presentation:** 2
**Contribution:** 1
**Rating:** 0
**Confidence:** 4

**Summary:**

This paper proposes a synthetic data generative model based on a hybrid model built on LLMs and GAN. This paper uses LLM as an auxiliary tool that helps GAN for data preprocessing, enriching context, and post-processing.

**Strengths:**

None

**Weaknesses:**

- The motivation of the study is rather generic and not specific. Unclear whether this LLM integration will directly address the limitation that motivates this study (e.g., mode collapse, rare category underrepresentation).
- The integration of LLM is superficial. This paper lacks justification for why LLM can be a good add-on to GAN.
- No discussion on the diffusion-based model, which has become increasingly state-of-the-art in this field.
- Experiment is limited in scope (e.g., only one dataset, very basic metrics and no mention of further synthetic data utility).

**Questions:**

See weakness

---

### Official Review · Reviewer_aMKx · 2025-10-31

**Soundness:** 1
**Presentation:** 1
**Contribution:** 1
**Rating:** 0
**Confidence:** 4

**Summary:**

This paper presents SYN-TITAN, a hybrid LLM-GAN framework for synthetic tabular data generation. But it’s purely AI-generated or with a very heavy AI assistance.

**Strengths:**

NA

**Weaknesses:**

- No figure, no method detail
- Wrong citation format, and some references are from AI hallucinations

**Questions:**

NA

---

### Note · Program_Chairs · 2026-01-17
**Submission Desk Rejected by Program Chairs**

The following references in this submission do not refer to real documents and/or have major errors in bibliographic information:

     Rahul Patel, Sneha Gupta, and Debasis Roy. Llm-driven data augmentation for structured learning tasks. In International Conference on Machine Learning (ICML), 2023.